# The Ecological Status and Change in High-Latitude Coral Assemblages at the Xuwen Coral Reef, Northern South China Sea: Insight into the Status and Causes in 2020

**DOI:** 10.3390/biology12020330

**Published:** 2023-02-17

**Authors:** Zhenxiong Yang, Wei Tao, Yue Liu, Wei Yu, Xiaojuan Peng, Chuqian Lu, Guangjia Jiang, Bin Chen, Wei Deng, Yihua Lv

**Affiliations:** 1South China Sea Environmental Monitoring Center, State Oceanic Administration, Guangzhou 510300, China; 2Nansha Islands Coral Reef Ecosystem National Observation and Research Station, Guangzhou 510300, China; 3Key Laboratory of Marine Environmental Survey Technology and Application, Ministry of Natural Resources, Guangzhou 510300, China; 4Marine Development Research and Promotion Center of Shenzhen, Shenzhen 518052, China

**Keywords:** coral community, environmental factors, anthropogenic activity, Xuwen coral reef

## Abstract

**Simple Summary:**

The Xuwen coral reef, as a potential refuge for corals to adapt to high-temperature migration, is an ideal “laboratory” to determine the impact of human activities on coral reef ecology in the northern South China Sea (NSCS). Global warming, coastal eutrophication and the decrease in herbivores has influenced coral reef degradation; compared with human activities, global climate change is the stronger influence on coral reef degradation on a large scale. For the Xuwen coral reef, which is in the coastal zone, anthropogenic stressors are more significant to the rapid degradation of its coral reef ecosystems. Under these circumstances, more evidence is required to understand the response of high-latitude reefs under natural and anthropogenic stresses. In this study, we investigated the ecological features of the scleractinian coral assemblage and their correlation with environmental factors on the Xuwen coral reef in 2020. Based on the data acquired by investigation and the implementation of statistical analysis methods, potential environmental conditions favoring coral communities in Xuwen were summarized. These results of the survey are expected to provide a basis for bridging the gap between economic development and sustainable management.

**Abstract:**

Taking the coral communities of the Xuwen coral reef in 2020 as the research object, we analyzed the species composition, diversity, and interspecific Spearman correlation of the scleractinian coral communities, investigated the features and spatial distribution of the scleractinian coral community, and discussed the correlation between the community composition and environmental factors to identify the affecting factors and their sources. These results showed that (1) compared with the survey in 2004, the coverage of corals in 2020 had significantly decreased, while the dominant genera were still *Goniopora* and *Porites*. The coral morphology was massive, and the diversity of the coral community (Shannon–Wiener index, *H’*) was 2.87. The distribution of coral was uneven. The competition among some dominant species of coral was intense. (2) The mass coral bleaching event in the NSCS in August 2020 did not cause severe coral death in the short term on the Xuwen coral reef. (3) The growth of the coral community in 2020 might be greatly affected by high suspended solids and nutrient levels, which were related to the current, mariculture, and coastal erosion. (4) Anthropogenic activities such as coastal aquaculture and fishing were the major factors leading to the reduction in coral coverage on Xuwen coral reef in the past 10 years.

## 1. Introduction

The Xuwen coral reef is located along the coast of Xuwen County, Leizhou Peninsula, Zhanjiang, Guangdong Province. The ^14^C age and uranium series ages of the Xuwen coral reef during the reef-forming period are about 7200 aBP [1,2,3], making it the only modern coral reef distributed along the coast of the Chinese mainland [4]. The Xuwen coral reef is also the largest coastal coral region in the Chinese mainland and is the northernmost coral reef in the world [5]. This subtropical and high-latitude coral area in the NSCS has been viewed as a potential “refuge” (e.g., Weizhou Island, Taiwan Island and Hong Kong) [6]. Global warming, eutrophication in coastal waters and the reduction in herbivores are often considered to be the main causes of coral reef degradation [7]. Compared with the impact of the global climate, the impact of anthropogenic activities is small, but in some local areas, once the interference of anthropogenic activities exceeds the tolerance limit of coral reef ecosystems, the ecosystems can degrade or even collapse [8].

Most large-scale coral reef ecological surveys in Xuwen began in 2000. Wang et al. reported eight families and 25 species of scleractinian corals from the Dengloujiao in 2000 [9,10]. Lu et al. [11] reported 35 species and 11 families of scleractinian corals in 2001. Coral reefs were distributed in narrow strips along the north-south coast, where scleractinian corals were mainly distributed at depths less than 8 m. Acroporidae, Poritidae and Dendrophylliidae were the most common families of scleractinian corals in Xuwen. Zhao et al. [12] reported four orders, 14 families and 34 species of scleractinian corals in 2004. Huang et al. [13] reported three orders, 18 families and 65 species of scleractinian corals in 2004, and the mean live coral cover was 28.9% [5]. Huang et al. [14] reported 57 species, 25 genera and nine families in 2008 [14], but the mean live coral cover was only 12.1%. Coral reefs were distributed in patches, mostly near the slope. The mean living coral cover in 2015 was 15.1%, with a range of 1.7–44.3% [15]. Li et al. [16] reported 46 species in 12 families and 16 newly recorded species in 2018. The mean coral cover at the 2 m water depth was 10.71%, and that at the 4 m water depth was 8.35%. According to previous reviews of the Xuwen coral reef [13], there were 75 species in 11 families (five undetermined species) of scleractinian corals.

Coral assemblages have declined significantly due to anthropogenic stressors and climate change in the last few decades [17,18]. On the Great Barrier Reef, it declined from 28.0 to 13.8% over 1985–2012 (0.53% year^−1^), a loss of 50.7% of the initial coral cover [19]. The coral reefs of the SCS have suffered a dramatic decline over the past 50 years, reflected in the decrease of live coral cover [18]. For example, living coral cover in Daya Bay declined from 76.6% to 15.3% from 1983 to 2008 [20]; off Yongxing Island, living coral cover decreased from 90% in 1980 to 10% in 2008 [21,22]. Intense anthropogenic activities are commonly thought to be the main factor that has driven the observed declines [18]. Escalating human activities negatively influence the survival of corals [17]. Pandolfi et al. [23] argued that human impacts began centuries ago and have followed a similar progression almost everywhere: the severe depletion and local extinction of megafauna preceded declines in fish and more recent widespread losses of corals, which have accelerated especially in the 1970s and 1980s. As reef corals are very sensitive to subtle changes of environmental conditions, they are widely used in the monitoring of seawater quality [18]. Although some ecology of scleractinian corals has been studied in Xuwen [5,6,8,9,10,11,12,13,14,15,16], with the rapid development of the economy and society, the influence of anthropogenic activities (e.g., inshore aquaculture, coastal engineering) is more intense than in the past. However, the effects of human activities in scleractinian coral assemblages have not been comprehensively and consistently studied in previous studies at NSCS, making it impossible to quantify and evaluate the impact of anthropogenic factors [24].

Under these circumstances, more evidence is required to understand the response or suitability of the high-latitude reefs under natural and anthropogenic stresses [25,26]. Especially after 2011, there were few large-scale ecological surveys of the Xuwen coral reef. Therefore, some important questions are present: (1) What is the ecological status of the Xuwen coral reef today and the changes that it is currently undergoing? (2) What cause-and-effect relationships are leading to the rapid decline of coral in the short run? (3) What is the cause of those environmental factors?

To answer these, a comprehensive study was carried out on the Xuwen coral reef in 2020. Based on the data acquired by investigation and the implementation of statistical analysis methods, potential environmental influence factors for coral communities on Xuwen coral reef were proposed. The results are expected to provide a basis for bridging the gap between economic development and the sustainable management of the Xuwen coral reef.

## 2. Materials and Methods

### 2.1. Field Surveys

The Xuwen coral reef (20.2°–20.5° N, 109.8°–109.9° E) is located in the NSCS (Figure 1), which is in a relatively high, subtropical latitude where monsoons, ambient sunlight and rainfall are important climatic characteristics. The mean annual air temperature is 24.5 °C; the lowest monthly mean temperature occurs in January (14.8 °C), the highest in July (33.2 °C). The mean annual precipitation is 1395.3 mm [27]. The mean annual solar radiation is 118.7 kcal/cm^2^ [5]. In summer, from July 2010 to July 2020, our monitoring results (unpublished data) showed that the mean annual water temperature ranged from 30.20 to 32.29 °C, the mean annual salinity ranged from 26.94 to 36.14‰, and the mean annual pH ranged from 8.04 to 8.27. Moreover, there is cyclonic circulation all the year round. With such climatic attributes and hydrological conditions, the NSCS provides an ideal habitat for coral growth and reef accretion.

Using video transect [28] and photo-quadrat methods [29], a field survey was implemented at the Xuwen coral reef in July 2020. Based on GPS positioning, 16 sections (XW1-XW14, XW 16, XW20, the locations of all the sampling sites are shown in Figure 1) were set up around the reef. In each section, two transects with measuring tapes (50 m long) were laid parallel to the shoreline at depths between 2 and 4 m (the depths of sections ranged from 1.5 to 6.7 m, those depths covered the area from the upper to lower limits of coral growth in our survey). For fish, the width of the transect each side was 2.5 m; for invertebrates, the width of the transect each side was 1 m. Ten quadrats (0.5 m × 0.5 m) were surveyed on each transect. Four photographs were taken for each quadrat, each occupying one-quarter of the quadrat. There were 23 transects and 230 quadrats in our survey.

The species identification of coral was based on close-up photographs, using the taxonomic framework developed by Veron [30] and Zou [31]; in this study, named species on the Xuwen coral reef mainly referred to the study of Huang et al. [5]. The number of species (species richness) was determined by summing up the number of species identified. Living coral cover was based on the line intercept transect procedure [32], which measured the lengths of corals that intercepted a transect line and assessed the percentage cover of the corals by their relative lengths [33]. In our survey, we measured the number of 50 m long transect at each 10 cm scale for corals. For corals within each radial transect, all corals ≥5 cm in diameter were identified to species (or genus for some smaller corals), including alive, bleached and recently dead corals. The mortality rate of coral was calculated as the percentage of recently dead corals (colonies entirely covered by turfing algae, but with discernible skeletal structure). We recorded the number and species of fish through video and close-up photographs. We extracted data such as sediment type, scleractinian coral assemblage, abundance of juvenile corals, density of coral reef fish, and macro-algae cover from the video transects and photo-quadrats.

The suitable habitat of coral was defined as the area from the upper to lower limits of coral growth by scuba diving, which was calculated by the geometry area calculation function of ArcMap software, using China Geodetic Coordinate System 2000, with Gauss Kruger 3 Degree projection, the central meridian was 111° E.

In addition, water sampling was performed once in 16 sites, including surface (0.5 m) and bottom (Water layer 2 m from the bottom) layers. The samplings of water parameters were temperature (T), salinity (S), pH, transparency, suspended solids (SS), dissolved oxygen (DO), chemical oxygen demand (COD), dissolved inorganic nitrogen (DIN, e.g., NH_4_^+^-N, NO_2_^−^-N, NO_3_^−^-N), inorganic phosphate (PO_4_^3−^-P), total nitrogen-(TN), total phosphorus (TP), oils and Chla. There were 405 water samples in our survey. Sample collection, storage, and transportation were conducted according to the Specification for Oceanographic Survey (GB/T 12763-2007) [34]. Briefly, water samples obtained with Go-Flo samplers were filtered through acetate membrane filters (0.45 μm pore sizes) and transferred into glass bottles and stored at 4 °C until analysis. Several important parameters in water (T, S, pH, transparency, SS, DO, COD, dissolved inorganic nitrogen, inorganic phosphate, TN, TP, oils and Chla) were determined according to the Specification for Marine Monitoring (GB 17378-2007) [35]. The seawater temperature and salinity were tested with a thermohalimeter. The pH values were determined with a pH meter. The SS samples were dried and weighed to measure the amount. The DO values were tested using the Winkler titration method. The COD values were measured using the alkaline potassium permanganate method. The values of NO_2_^−^-N and NO_3_^−^-N, PO_4_^3−^-P, TN, and TP were analyzed using the flow analysis method. The NH_4_^+^-N values were determined using the hypobromite oxidation method. The oils values were analyzed using the ultraviolet spectrophotometry method. The Chla values were tested using the fluorescence spectrophotometry method.

### 2.2. Data Extraction and Treatment

The importance values (IVs) [36] of various corals were calculated to analyze the assemblage structure and the dominant species of the corals. The species with the highest IVs was identified as dominant species, and the species with the second highest IVs was defined as a codominant species [37,38,39,40]. The IVs were used to evaluate the dominant species as follows:Relative abundance *RA_i_* =*n_i_*∕∑*n_i_*,(1)
Relative coverage *RC_i_* =*c*_i_∕∑*c*_i_(2)
Relative frequency *RF_i_* =*f_i_*∕∑*f_i_*
(3)
Importance value *IVi* = *RA_i_* + *RC_i_* + *RF_i_*(4)
where *n_i_* is the number of colonies of the coral species *i* in each quadrat, *c_i_* is the cover of the coral species *i* in each quadrat, *f_i_* is the frequency of the coral species *i* in each quadrat, *RA_i_* is relative abundance, *RC_i_* is relative coverage, and *RF_i_* is relative frequency, respectively.

The species diversity (Shannon–Wiener index, *H’*), evenness (Pielou index, *J’*) and species richness (Margalef index, *D*) were used to analyze the diversity characteristics of coral communities as follows [40,41]:(5)$$H^{\prime }=-OR\;H=-\sum \left[\left(P_{i}\right)\times \ln\left(P_{i}\right)\right]\;$$
*J′* = *H′*/ln*S*(6)
*D* = (*S* − 1)/ln*N*(7)
where *P_i_* is the proportion of the coverage of species *I* in the total coverage of the community, *S* is the total number of species in the transect, and *N* is the coverage of total number of individuals.

We adopted the method of Spearman rank, correlation coefficient and significance methods to analyze the relationships among the dominant coral species [42]. Cluster analysis was used to preliminarily establish the classification relationship of coral community distribution in different environmental factors; principal component analysis (PCA) was used to identify the major environmental factors among different sites; Pearson correlation (2-tailed) analysis was used to analyze the relationship among the richness, densities, frequencies, covers, etc.; and linear regression was used to fit the regression equation among the assemblage structure and environmental factors. Redundancy analysis (RDA) was used to analyze the relationship among dominant genera, dominant species, and environmental factors of coral. We determined whether they had a linear relationship by detrended correspondence analysis (DCA). If the values of all ranking axes in DCA are less than 3, RDA was selected. The species matrix was converted by Lg (x + 1), and the environmental data were converted by Lg (x + 1) except pH. Cluster analysis was implemented in Primer 5.0 software. Pearson correlation analysis and linear regression were performed using SPSS 16 software. The DCA, PCA, and RDA were conducted in Canoco 4.5 software [43]. Figures were generated using ArcGIS 10.7.

## 3. Results

### 3.1. Coral Community Structure

The records of species at the Xuwen coral reef are shown in Table 1. A total of eight families, 19 genera, and 33 coral species were identified at the Xuwen coral reef during our ecological survey in 2020. Most species belonged to the family Merulinidae (54.4%), followed by Poritidae (24.2%). *Dipsastraea* was the genus with the most species present.

We calculated that the suitable habitat of coral was approximately 14.61 km^2^ in 2020, and living coral cover ranged from 5.2% to 26.1%, with an average of 12.1%. The corals of Xuwen coral reef had an uneven spatial distribution, were mainly distributed within 5.0 m depth, and the highest percent of living coral cover was near Shuiwei and Fangpo (Figure 2). At 2 m, the living coral cover ranged from 1.8% to 20.0%, and the average living coral cover was 10.5%. At 4 m, the living coral cover ranged from 6.0% to 39.0%, and the average living coral cover was 16.7%. 

According to importance values (IVs), the dominant coral species, genera, and families are shown in Table 2. *Goniopora gracilis* (11.6% IVs) was the most dominant coral species, and *Bernardpora stutchburyi* (8.7% IVs), *Montipora turgescens* (8.0% IVs), *Turbinaria peltata* (7.9% IVs), and *Porites lutea* (7.7% IVs) were codominant species. 

The spearman rank correlation between the dominant species in coral communities is shown in Figure 3. The number of positive correlations (46, accounts for 59.0%) between the dominant species was slightly higher than those of negative correlations (32, accounts for 41.0%), and the number of significant correlations (*p* < 0.05) was 6, accounts for 7.7%, indicating that the interspecific association of the coral community was relatively loose. On the other hand, some species were highly correlated, such as the significant positive correlation between *Favites abdita* and *Platygyra daedalea*, and between *Porites lutea* and *Goniopora* sp. (*p* < 0.01, Figure 3).

The Shannon–Wiener index (*H’*, Figure 4) ranged from 1.59 to 3.69, with an average of 2.87. The *H’* varied greatly at different water depths. At 2 m, the range of *H’* was 1.59~3.39, with an average of 2.65 (at 25.0% of the sites), and the sites with H’ > 3 were the XW4, XW13 and XW20 sites. At 4 m, the range of *H’* was 2.40~3.69, with an average of 2.85. A total of 8.6% of sites with *H’* > 3 were the XW8 and XW10 sites.

The mean abundance of juvenile corals (largest diameter < 4 cm) was 3.01 ind./m^2^. Juvenile corals were mainly distributed at the XW1 and XW14 sites. Most species of juvenile corals were in the Merulinidae family, which was the same as the overall dominant species. The mean mortality rate of recently dead corals was 2.8%. The site with the highest mortality rate was the XW1 site, which was close to Liusha Bay, where living coral cover was also the lowest (5.2%). A total of 62.5% of the sites were found to have coral bleaching, and the highest was at the XW16 site, and the mean coral bleaching rate was 1.60% (in July 2020) (Table 3).

### 3.2. Reef Dwelling Organism Composition and Environmental Factors in Water

The dominant macroalgal species were *Caulerpa racemosa*, *Caulerpa sertularioides*, etc. The height of macroalgae ranged from 18 cm to 20 cm. The mean macroalgae cover was 0.7%, and the highest was near Liusha Bay (in the XW1 site, which was also the lowest living coral cover) and Dengloujiao (in the XW13 site). There were six families and seven species of coral reef fish, mainly of the Pomacentridae family. The dominant species was *Neopomacentrus bankieri*. The mean density of coral reef fish was 8.38 ind./250 m^2^, and 96.7% of coral reef fish were less than 10 cm in length. The mean density of invertebrates was 155.06 ind./100 m^2^. The dominant group was sponges, whose density accounted for 59.73% (Table 3). The mean reef cover was 46.2%, and 37.5% of the survey sites had a mean reef cover greater than 50%. The surface sediments of the reefs were mainly sand, with a mean grain content (%) of 74.1%. 

Approximately 56% of the sites were seriously sandy. The mean suspended solids concentration was 17.0 mg/L, and approximately 68.8% of the sites exceeded 10 mg/L (Table 4).

### 3.3. Water Environment Factors Affecting Coral Growth

Both the CLUSTER tests (d = 21, approximately 89% of similarity) and PCA tests (the explained proportion was 84.2%, PCA1 = 35.1%, PCA2 = 27.1 %) are shown in Figure 5. The habitat zone of scleractinian coral was divided into two categories. One was the XW6, XW7 and XW16 sites, which were near Dongchang Bay and Baigong. These sites, suffered the greatest impact of human activity, had the strongest relationship with coral bleaching (BR, it was 8.2% at the XW16 site) and suspended solids (SS, mean SS > 30 mg/L), while other sites suffered from human activity were the habitats of other survey sites (except the XW6, XW7 and XW16 sites). The highest living coral cover (26.1%) and the largest number of species (19) were in the XW8 site, which was highly positively correlated with transparency (TRAN) and reef coverage (RC). In addition, it should be noted that XW8 site was closely to XW6 site, where the concentration of suspended solids was 31.5 mg/L, however, corals were growing there. The water environment of site XW16 was the same as that of site XW8, but the concentration of Chla was higher than that at site XW8 (XW16, 4.58 µg/L; XW8, 8.69 µg/L).

Correlation and multiple linear regressions were used to analyze the relationship between ecological features and major environmental factors (Table 4). The RDA method was adopted to analyze the relationship among the dominant genera, species of coral, and environmental factors (DCA tests for dominant coral genera showed that the length of the first ranking axis of DCA was 1.314, and the values of all ranking axes were no more than 1.314).

Our results are shown in Table 5 and Figure 6. There was a significant negative correlation between living coral cover and NH_4_^+^-N (-0.561, *p* < 0.05). For dominant genera, the significant environmental impact factors (by Monte Carlo test; the proportion of interpretation was 86.5%) were nutritional salt (mainly NH_4_^+^-N, *p* = 0.002) and suspended solids (*p* = 0.022). For dominant species, the significant environmental impact factor (by Monte Carlo test; the proportion of interpretation was 68.8%) was nutritional salt (mainly NH_4_^+^-N, *p* = 0.002). At the genus level, *Goniopora* and macroalgae (mainly *Caulerpa racemosa*) had formed a large competitive relationship (especially in XW1). *Goniopora gracilis* and *Bernardpora stutchburyi* were positively correlated with NH_4_^+^-N, which indicated the possibility that these corals could grow well in water with high NH_4_^+^-N content. The number of coral species was negatively correlated with chlorophyll a (Chla, −0.751, *p* < 0.01). The coral diversity index (*H’*) was significantly negatively correlated with Chla (−0.736, *p* < 0.05). 

## 4. Discussion

### 4.1. Temporal Variations in the Coral Community on the Xuwen Coral Reef

Past research [5,6,8,9,10,11,12,13,14,15,16] showed that the living coral cover of the Xuwen coral reef had been greatly reduced in recent years, and in 2020, was reduced by 56.4% compared to that in 2004 (Table 6). Within the past few decades, on fringing reefs along the Chinese mainland and around Hainan Island, the amount of living coral cover has declined from 60% to 20% [44]. For instance, off Sanya Luhuitou, living coral cover decreased from 80–90% in 1960 to 12% in 2009 [20,45]; off Weizhou Island, living coral cover decreased from 60–80% in 1991 to 8–18% in 2010 [24]. In addition to gathering information about changes in living coral cover, quantifying changes in coral taxonomic assemblage structure is also critical for measuring ecosystem state [46]. The dominant coral groups were the same as the findings of Zhao et al. [47], including Poritidae, Merulinidae, and Dendrophylliidae. Moreover, *Goniopora* sp. was the dominant species from 2004 to 2020. These dominant species often grew slowly and are distributed widely, which were features of a unique northern coral reef ecosystem [5,48]. These massive and encrusting corals have a higher capacity to moderate environmental stress with their higher coral tissue thickness and low growth rate [49]. The results showed that the average values of Shannon–Wiener index (*H’*) was 2.87, species distribution in the coral community was uneven, and there was still a trend of population differentiation and succession. The reason may be that human activities (e.g., nutrient input) have disturbed the mechanism of interspecific niche complementarity in recent years, weakening the positive effect of species diversity on community stability [50], suggesting that some species with strong tolerance to environmental disturbance restrained the growth of other species. The degradation of the assemblage structure on the Xuwen coral reef was the same as that in other global coral reefs. Sancia et al. [51] investigated the effects of prolonged stress on the species composition of reefs in Jakata Bay, in which case, prolonged stress resulted in the loss of species belonging to Acroporidae, Milleporidae, and to a lesser extent Poritidae. A study in the Netherlands Antilles showed that rare coral species became extinct in permanent quadrats and that other species became less abundant over a 20-year period, likely as a result of urbanization [52]. The coral assemblage structure had undergone degradation, with the dominant group shifting from high complexity branching, foliaceous and massive colonies to a simpler group of massive morphologies on Weizhou Island [39].

### 4.2. Global Warming Affecting the Xuwen Coral Reef

The habitat fragmentation of the Xuwen coral reef was significant. It should be noted that naturel disturbances may play a more visible role in the dynamics of coral assemblages in small areas than in larger areas [53]. For most in-shore warm water corals, water temperature is importance for their growth [54]. Water temperature ranging from 20 °C to 30 °C is most suitable for most coral growth, while the beneficial effects are suppressed when the temperature is <20 °C or >30 °C. Yang et al. [55] reported the most suitable growth temperature of the *Porites lutea* and *Galaxea fascicularis* on the Xuwen coral reef, which was 29.5 °C. During this study, the water temperature was 31.75 °C–32.70 °C, which was higher than the suitable temperature of most corals. 

The World Meteorological Organization “Global Climate Conditions 2020” [56] had reported that 2020 was one of the three warmest years on records, and the global average temperature today is approximately 1.2 °C higher than that of preindustrial times. Based on the projected future heat stress used by the Intergovernmental Panel on Climate Change (IPCC), under RCP 8.5 (CO_2_ emissions continue to rise as usual), two severe bleaching events per decade will be familiar scenarios at 25 of the 29 World Heritage reefs (86%) by 2040 [57]. In previous studies, Chen et al. [58] speculated that Xuwen would enter the middle tropical zone (leaving its current position in the north tropics) starting in 2020. Yu et al. [59] also found that there was a significant upward trend in SST over four decades in Lei-Qiong District. 

From July 2020 to July 2021, a mass coral bleaching event was observed. In July 2020 (16 sites, the mean mortality rate of coral in one year was 2.8% and the mean living coral cover was 12.1%, this study), 62.5% of sites experienced coral bleaching, with 1.6% mean bleaching. In August 2020, a mass coral bleaching event was recorded in NSCS [60]. Ten sites and 26 transects were set up in Xuwen coral reef, in which all sites had been found coral bleaching, ranged from 68.0% to 100.0% with increasing sea temperature, while mean coral bleaching reached 89.3%. However, in the later period of monitoring, as the sea temperature fell, coral bleaching decreased, with a mean of 5.8% in July 2021 (17 sites, the mean mortality rate of coral in one year was 0.33% and the mean living coral cover was 11.7%, unpublished data). 

Lyu et al. [60] also reported this coral bleaching event on Weizhou Island from August to September in 2020, which was similar to the coral bleaching in Xuwen coral reef. In their study [60], an undersea monitoring system on Weizhou Island was used to analyze image capture and evidently showed that the coral colony turned pale and began to undergo sporadic bleaching in late May 2020. In mid-September, the SST began to decline gradually below the bleaching threshold, and the heat stress weakened. Later, photos captured by the monitoring system revealed signs of bleached coral recovery through 31 October. The follow-up survey showed that in situ areas experienced no loss of coral cover during the previous year’s mass bleaching event, but a falling bleaching percentage was observed, which is surprising since the coral was experiencing the same level of thermal stress.

It is suggested that the reason why global warming failed to cause severe coral death in the short term on the Xuwen coral reef was more likely the unique features of ocean currents [47]. There is no large land-based runoff nearby the Xuwen coral reef. The water temperature of the Xuwen coral reef in summer was little different from that of the coastal water with high temperature and low salt [61]. Our monitoring data showed that the mean annual water temperature ranged from 30.20 to 32.29 °C in the last ten years in summer. Zhao et al. [3,47] estimated that the Qiongzhou Strait tidal current was strong and that the water was exchanging and mixing frequently between Beibu bay and the NSCS, which could make the sea temperature low in the summer. This may be one reason that corals on the Xuwen coral reef were still growing or recovering even though the sea temperature was high for a period in 2020.

### 4.3. Escalating Anthropogenic Activities Weakening the Coral Habitat Function on the Xuwen Coral Reef

The major human-induced stressors on coral reefs have been known for decades, including sewage, siltation, industrial discharge, urban development, and destructive fishing methods [62]. Local anthropogenic influences might determine whether subtropical coastal water can serve as a refugia to protect coral habitats from warming [63]. Wang et al. [64] reported that the living coral cover in Fangpo village, decreased from 30–40% to 10% in 2000–2004, which was at least in part due to the disorderly cultivation of pearl oysters on a large scale. Huang et al. [5] found that after the Demarcation and Fishery Cooperation Agreement between China and Vietnam had taken effect starting in 2010, 32,000 square kilometers of high-quality Chinese fishing grounds had been degraded, which prompted 6600 high-power fishing boats to withdraw from the traditional fishing area. Many Chinese fishermen transferred to the near shore, which brought tremendous fishing pressure on the coast of the Beibu Gulf. There were a large number of fixed nets and waste fishing nets observed in our survey (Figure 7), which were mainly distributed in the core zone of Shuiwei (XW2~XW4 sites), Dongchang Port-Baoxi Port experimental zone (XW6~XW8 sites), and some shallow water areas (XW14 site). These results showed that human activities nearshore, such as mariculture and fishing, were frequent on the Xuwen coral reef. Indeed, damage from extreme weather, such as typhoons, can directly damage reefs and cause coral mortality. Two typhoons (Super Typhoon Rammasun, No. 1409; Typhoon Kalmaegi, No. 1415) struck Xuwen coral reef in 2014. From 2015 to 2018, without typhoons stricking Xuwen coral reef directly, the mean living coral cover in 2015 was 15.1% [15] and in 2018 was 8.96% [16], it’s suggested that the coral did not recover naturally. Therefore, in addition to extreme weather damage such as typhoons, human activities also changed the Xuwen coral community significantly in recent years.

### 4.4. Factors Influencing Coral Community Structure

Most corals at the Xuwen coral reef were distributed in shallow water, where the concentrations of suspended solids were high in our survey. In these areas with high concentrations of inorganic nitrogen and phosphate in the local area, macroalgae flourished. Our results showed that there was a significant negative correlation between live coral cover and NH_4_^+^-N (−0.561, *p* < 0.05). The number of coral species was negatively correlated with chlorophyll a (Chla, −0.751, *p* < 0.01). The coral diversity index (*H’*) was significantly negatively correlated with Chla (−0.736, *p* < 0.05). The high Chla content indicated high phytoplankton biomass, which suggested that there was a high nutrient content in the water. In addition, the fish density was positively related to the abundance and diversity of coral species and the most dominant genera. High fish density (especially phytophagous fish) can affect macroalgal growth, which is conducive to coral growth. Based on these statistical results, it is suspected that the growth of the coral community in 2020 might have been mostly affected by suspended solids and nutrients.

As mentioned above, the environment was almost identical between the XW16 and XW8 sites, though XW16 site had a higher concentration of Chla and sand coverage. It might indicate XW16 site was not suitable for coral growth or recovery.

#### 4.4.1. High Suspended Solids on the Xuwen Coral Reef

The seawater of Xuwen coral reef was turbid. The average concentration of suspended solids on the Xuwen coral reef was 17.0 mg/L, where approximately 68.8% of survey sites had an average concentration of more than 10 mg/L. Elevated turbidity can reduce light, thus affecting the photosynthesis of zooxanthellae [39]. Suspended matter and sediments have a negative impact on coral larval survival, settlement rate and growth rate. In addition to corals, increases in sedimentation can affect other reef-associated organisms [65]. Settlement of early-life stages of seaweeds is disrupted by high levels of sediment [66].

The statistical results in 2020 showed that the suspended solids were significantly positively correlated with coral bleaching (0.567, *p*< 0.05), which was consistent with the results of Wang et al. [67]. The capacity of corals to remove suspended solids in water is not only closely related to the species, morphology, living habits and even growth direction of corals, but is also affected by the suspended solids concentration [68], e.g., *Porites* sp. can strongly resist to suspended solids in water [69]. *Acropora* sp. is sensitive to the content of suspended solids in water [70], and it is easier to bleach than massive corals such as *Porites* sp. [71]. *Favites* sp. has a larger coral cup and higher heterotrophic capacity (to capture organic particles as food), and resists high suspended solids content more strongly [72]. Our results also showed the dominant species of corals (e.g., *Dipsastraea*, *Goniopora* and *Favites*) were massive corals, all of which can grow well in water with high level of suspended solids.

It is possible the high suspended solids content is the result of the hydrodynamic conditions, coastal erosion, and traditional salt production in Xuwen. The frequent exchange and mixing of seawater and coastal water outside the Qiongzhou Strait [3], the turbulent diffusion caused by the upwelling, and wind waves were the driving forces of sediment diffusion and shoreline erosion in Xuwen [73]. In addition, high suspended solids content was observed near Baigong in this study (XW7, XW8, XW16 had suspended solids content > 30 mg/ L), which was not only seriously eroded by the sandy shoreline but also by the traditional sea-salt production in Xuwen [74].

#### 4.4.2. Elevated Nutrient Loads Affect Coral Growth and Reproduction

The nutrients in tropical coral reefs are generally poor: the nitrate (NO_3_^−^-N) is 0.1–0.5 μmol/L, the ammonium salt (NH_4_^+^-N) 0.2–0.5 μmol/L, and the inorganic phosphorus less than 0.3 μmol/L [75]. In this study, the average concentration of the dissolved inorganic nitrogen (DIN) was 3.66 μmol/L (2.79 μmol/L in 2006 [76]), 1.52 μmol/L for NH_4_^+^-N (0.79 μmol/L in 2006 [76]), 1.79 μmol/L for NO_3_^−^-N (1.57μmol/L in 2006 [76]), all of which were higher than the concentrations found in tropical coral reefs and exceeded eutrophication threshold levels of nutrients (DIN = 1 μmol/L) [77]. Eutrophication may negatively affect coral biodiversity and coral cover in coastal coral reefs [78,79], and unfavorable ratios of N/P in seawater can alter the composition of the photosynthetic membranes of zooxanthellae, resulting in an increased susceptibility of corals to bleaching [80]. Thurber et al. [81] presented a long-term nutrient enrichment experimental evidence that coastal nutrient loading was one of the major factors contributing to the increasing levels of both coral disease and coral bleaching. In this study, it was also found that the N/P range was 15–77 and that the mean value was 27. The N/P of sites the XW6–XW7 (high chlorophyll a, content of 17.13–19.36 µg/L), XW13–XW14, XW8–XW9 and XW16 sites were more than 30, suggesting phosphorus limitation.

Our results showed that NH_4_^+^-N values were negatively correlated with most coral communities, which was similar to the results of the previous study [82]. After long-term (14–21 days) exposure to high NH_4_^+^-N (20 μmol/L), the photosynthetic efficiency (F_v_/F_M_) of corals decreased significantly, the host protein content increased evidently, and the symbiotic pigmentation (Chla content) also had obviously increased. This NH_4_^+^-N content may mostly come from the excretion of organisms and the decomposition of organic matter. The DIN was transferred from the seawater to the sediment, participating in microbial N cycling, microalgal uptake and sedimentary adsorption [83]. The NO_3_^−^-N was the primary component of DIN during the daytime, which changed to NH_4_^+^-N during the nighttime [84]. Increasing oxygen demand in the elevated biomass by nutrients, which results in a lack of oxygen, killed coral quickly [84]. Our results demonstrated that the range of DO was 4.04–5.91 mg/L in 2020 (all sites exceeded the class Ⅰ seawater quality standard in China (>6 mg/L), and some sites (e.g., XW01) were located in Liusha Bay, with relatively poor water exchange. Our results showed that there have been many aquaculture activities in Xuwen coral reef recently. For example, the dynamic monitoring data of the sea area near Xuwen coral reef (unpublished data) showed that there were 116 registered projects near the Xuwen coral reef by 2020, with a total sea area of 5743.6942 hectares, mainly shellfish culture and fish cage culture. It is possible aquaculture activities cause eutrophication and lead to an anoxic or anaerobic state of the sediment [73], which might be the cause of the high correlation between NH_4_^+^-N and the coral community in this study.

## 5. Conclusions

Based on the ecological investigation of the Xuwen coral reef in July 2020, the ecological features and impact factors of scleractinian corals on Xuwen coral reef were analyzed. These results showed that compared with 2004, the area and coverage of corals were significantly decreased, and corals were distributed in patches, mainly in the core zone of the reserve. The dominant genera were still *Goniopora* and *Porites*. *Bernardpora stutchburyi*, *Turbinaria peltata* and *Plesiastrea versipora* were common dominant species on the Xuwen coral reef, which all showed resistance to interference. The coral morphology was massive, with a diversity of 2.87 of the coral community (Shannon–Wiener index, *H’*), which indicates the distribution of corals was loose.

The mass coral bleaching event happened in the NSCS in August 2020 did not cause severe coral death in the short term on the Xuwen coral reef. The comprehensive statistical results suggested that the coral community in 2020 might be greatly affected by high suspended solids and elevated nutrient levels, which were related to the current, mariculture and coastal erosion in and around Xuwen coral reef. Anthropogenic activities such as coastal aquaculture and fishing had been the major factors causing the reduction in coral coverage on Xuwen coral reef in the past 10 years.

## Figures and Tables

**Figure 1 biology-12-00330-f001:**
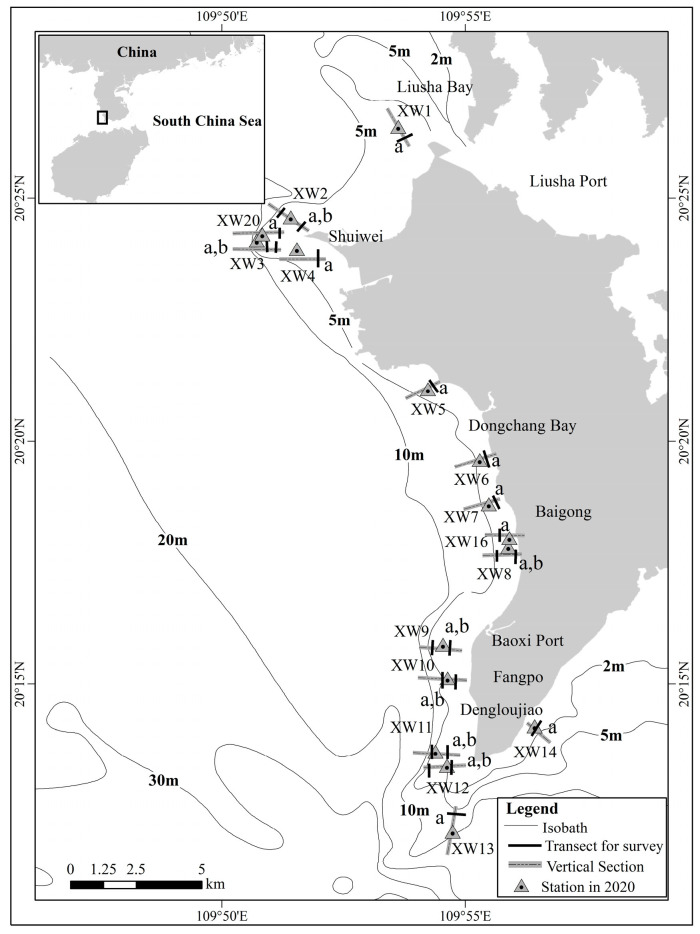
Location of the 2020 coral ecological assessment survey sites in Xuwen (a is 2 m transect, b is 4 m transect).

**Figure 2 biology-12-00330-f002:**
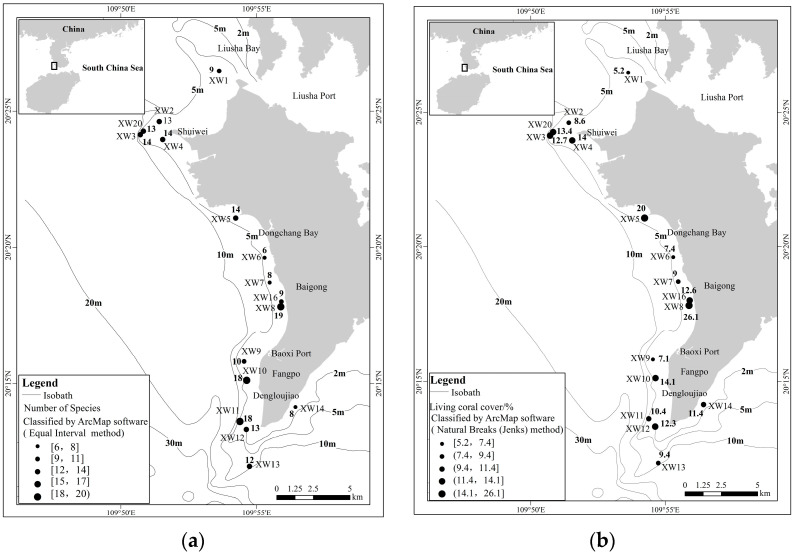
The distribution of scleractinian coral species number (**a**) and coverage (**b**).

**Figure 3 biology-12-00330-f003:**
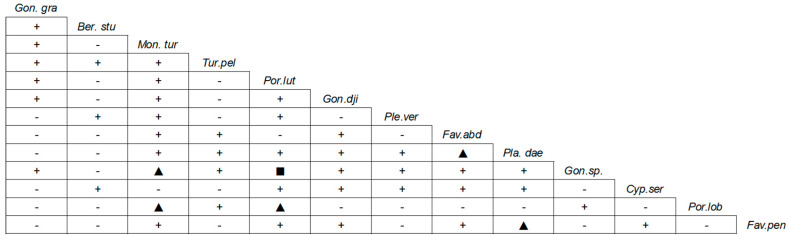
Spearman rank correlation analysis of the main species in coral communities. Positive correlation: ■. *p* < 0.01, ▲. *p* < 0.05, + *p* > 0.05; Negative correlation: - *p* > 0.05; No correlaton: O. *Gon.gra: Goniopora gracilis*; *Ber.stu: Bernardpora stutchburyi*; *Mon.tur: Montipora turgescens*; *Tur.pel: Turbinaria peltata*; *Por.lut: Porites lutea*; *Gon.dji: Goniopora djiboutiensis*; *Ple.ver: Plesiastrea versipora*; *Fav.abd: Favites abdita*; *Pla.dae: Platygyra daedalea*; *Gon.*sp.: *Goniopora* sp.; *Cyp.ser: Cyphastrea serailia*; *Por.lob: Porites lobata*; *Fav.pen: Favites pentagona*.

**Figure 4 biology-12-00330-f004:**
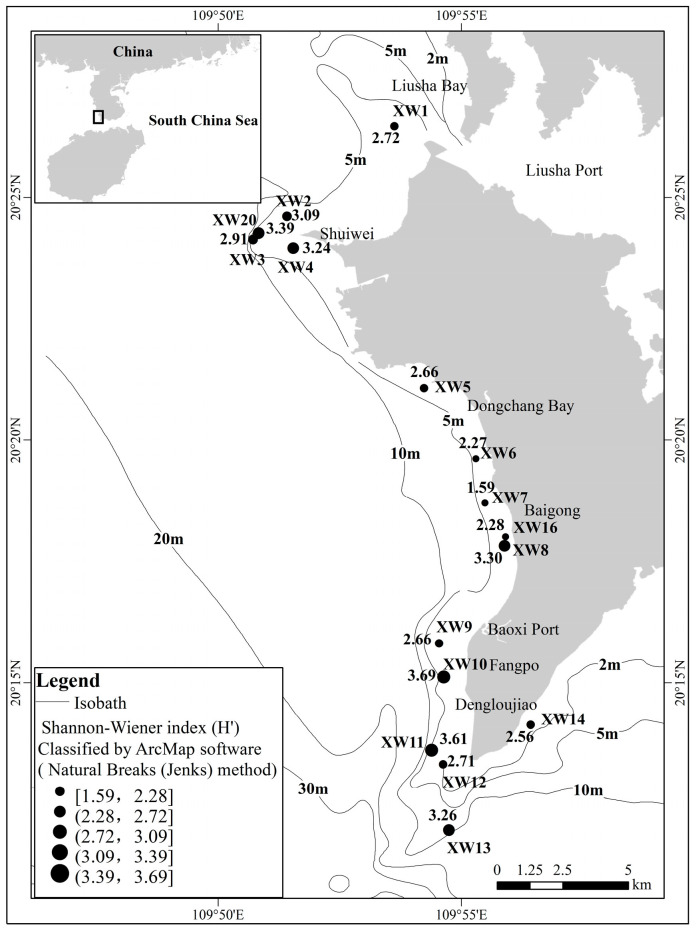
The diversity index of scleractinian coral on Xuwen coral reef.

**Figure 5 biology-12-00330-f005:**
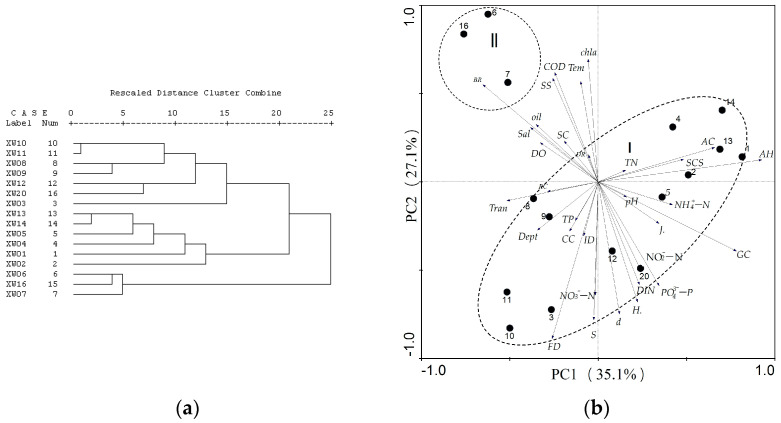
The Cluster and PCA of the ecological environment in the coral reef area. (**a**): Cluster; (**b**): PCA). Arrow indicates ecological environment factors, and the circle indicates the site; Dept: Depth; Tran: Transparency; Tem: Temperature; Sal: Salinity; COD: Chemical oxygen demand; SS: Suspended solids content; Oil: Oils; NO_3_^−^-N: Nitrate-NO_3_^−^-N; NO_2_^−^-N: Nitrate- NO_2_^−^-N; NH_4_^+^-N: Ammonium salt; DIN: Dissolved inorganic nitrogen; PO_4_^3−^-P: Inorganic phosphate; Chla: Chlorophyll-a; TP: Total phosphorus; TN: Total nitrogen; AC: Coverage of macroalgae; AH: Height of macroalgae; CC: Live coral cover; SCS: Scleractinian juvenile corals; FD: Density of fish; ID: Density of invertebrate; S:Number of coral species; BR: Coral bleaching rate; DR: Coral mortality; RC: Reef coverage; GC: Gravel coverage; SC: Sand coverage; *H’*: Shannon-Wiener index; *J’*: Pielou index; *D*: Simpson index.

**Figure 6 biology-12-00330-f006:**
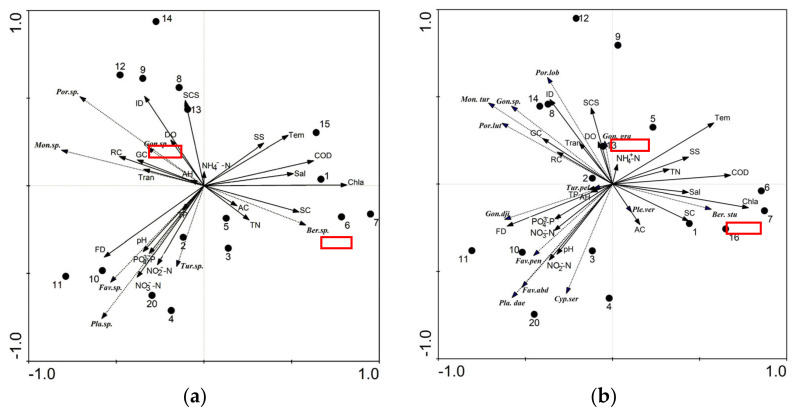
RDA analysis of environmental factors between dominant genera (**a**) and dominant species (**b**). Black dashed arrows indicate the corals genera (species), Latin abbreviations indicate the top several dominant genera (species), red boxes showed the 2 most dominant genera (species), black solid arrows indicated environmental factors and black circles indicate the site number.

**Figure 7 biology-12-00330-f007:**
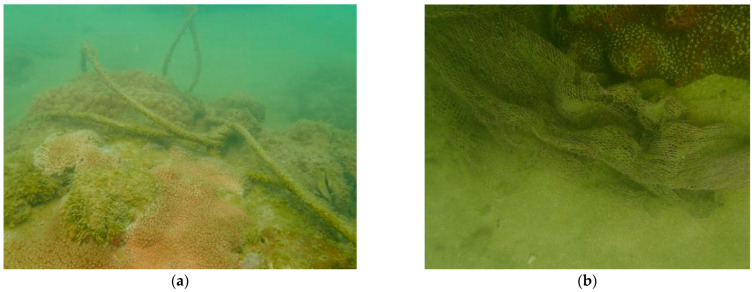
Coral covered by waste fishing nets in this survey (**a**) Rope; (**b**) Fishing net.

**Table 1 biology-12-00330-t001:** The records of species at the Xuwen coral reef in 2020.

Categories	Phylum/Family	Genus	Species	Dominant Species
Coral	Poritidae	*Bernardpora*	*Bernardpora stutchburyi*	a 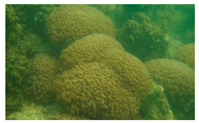 b 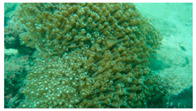 c 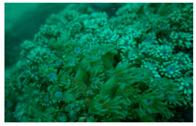 *Goniopora* sp. (a–c) d 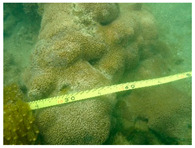 e 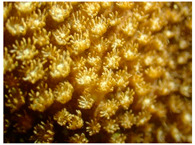 *Bernardpora stutchburyi* (d–e)
		*Goniopora*	*Goniopora columna*	
			*Goniopora djiboutiensis*	
			*Goniopora gracilis*	
			*Goniopora* sp.	
		*Porites*	*Porites lobata*	
			*Porites lutea*	
			*Porites* sp.	
	Agariciidae	*Pavona*	*Pavona explanulata*	
	Acroporidae	*Acropora*	*Acropora humilis*	
		*Montipora*	*Montipora turgescens*	
	Merulinidae	*Astrea*	*Astrea curta*	
		*Coelastrea*	*Coelastrea palauensis*	
		*Cyphastrea*	*Cyphastrea serailia*	
		*Dipsastraea*	*Dipsastraea favus*	
			*Dipsastraea matthaii*	
			*Dipsastraea rotumana*	
			*Dipsastraea* sp.	
			*Dipsastraea speciosa*	
		*Echinopora*	*Echinopora gemmacea*	
		*Favites*	*Favites abdita*	
			*Favites flexuosa*	
			*Favites halicora*	
			*Favites pentagona*	
		*Goniastrea*	*Goniastrea aspera*	
			*Goniastrea retiformis*	
		*Merulina*	*Merulina ampliata*	
		*Platygyra*	*Platygyra daedalea*	
			*Platygyra* sp.	
	Dendrophylliidae	*Turbinaria*	*Turbinaria peltata*	
	Siderastreidae	*Pseudosiderastrea*	*Pseudosiderastrea tayamai*	
	Plesiastreidae	*Plesiastrea*	*Plesiastrea versipora*	
	Lobophylliidae	*Lobophyllia*	*Lobophyllia corymbosa*	
Fish	Pomacentridae	*Neopomacentrus*	*Neopomacentrus bankieri*	f 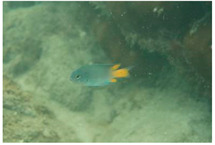 *Neopomacentrus bankieri* (f)
		*Abudefduf*	*Abudefduf bengalensis*	
	Pomadasyidae	*Diagramma*	*Diagramma pictum*	
	Lutjanidae	*Lutjanus*	*Lutjanus ehrenbergii*	
	Microdesmidae	*Parioglossus*	*Parioglossus philippinus*	
	Labridae	*Halichoeres*	*Halichoeres nigrescens*	
	Gerreidae	*Gerres oyena*	*Gerres oyena*	
Algae	Chlorophyta	*Boergesenia*	*Boergesenia forbesii*	g 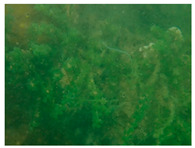 *Caulerpa racemosa* (g)
		*Bryopsis*	*Bryopsis pennata*	
		*Caulerpa*	*Caulerpa racemosa*	
			*Caulerpa sertularioides*	
	Ochrophyta	*Dictyota*	*Dictyota* sp.	
		*Lobophora*	*Lobophora variegata*	
		*Padina*	*Padina* sp.	
	Rhodophyta	*Ceratodictyon*	*Ceratodictyon spongiosum*	
		*Hydropuntia*	*Hydropuntia edulis*	
		*Agarophyton*	*Agarophyton tenuistipitatum*	
		*Jania*	*Jania adhaerens*	
		*Laurencia*	*Laurencia decumbens*	
		*Peyssonnelia*	*Peyssonnelia rubra*	
			*Peyssonnelia* sp.	

**Table 2 biology-12-00330-t002:** The dominant species of coral reefs at different water depths of the Xuwen coral reef.

Area	Family	Species	IVs	Percentage of IVs (%)
All	Poritidae	*Goniopora gracilis*	1.438	11.6
All	Poritidae	*Bernardpora stutchburyi*	1.088	8.7
All	Acroporidae	*Montipora turgescens*	0.998	8.0
All	Dendrophylliidae	*Turbinaria peltata*	0.978	7.9
All	Poritidae	*Porites lutea*	0.962	7.7
All	Merulinidae	*Favites abdita*	0.773	6.2
All	Plesiastreidae	*Plesiastrea versipora*	0.707	5.7
All	Poritidae	*Goniopora djiboutiensis*	0.549	4.4
2 m	Poritidae	*Goniopora gracilis*	1.401	11.9
2 m	Poritidae	*Bernardpora stutchburyi*	1.204	10.2
2 m	Dendrophylliidae	*Turbinaria peltata*	0.911	7.7
2 m	Poritidae	*Porites lutea*	0.894	7.6
2 m	Merulinidae	*Favites abdita*	0.844	7.1
2 m	Acroporidae	*Montipora turgescens*	0.810	6.9
2 m	Plesiastreidae	*Plesiastrea versipora*	0.794	6.7
4 m	Poritidae	*Goniopora gracilis*	1.476	10.8
4 m	Acroporidae	*Montipora turgescens*	1.340	9.8
4 m	Dendrophylliidae	*Turbinaria peltata*	1.129	8.2
4 m	Poritidae	*Porites lutea*	1.117	8.1
4 m	Poritidae	*Bernardpora stutchburyi*	0.867	6.3
4 m	Poritidae	*Goniopora djiboutiensis*	0.847	6.2
4 m	Poritidae	*Goniopora sp.*	0.787	5.7
4 m	Poritidae	*Porites lobata*	0.648	4.7

**Table 3 biology-12-00330-t003:** The main reef dwelling organism on Xuwen coral reef.

Items	Abundance of Juvenile Corals (ind./m^2^.)	Mortality Rate (Recently Dead Corals, %)	Bleaching Rate (%)	Macroalgae Cover (%)	Density of Fish (ind./250 m^2^)	Sponge Cover (%)	Density of Invertebrates (ind./100 m^2^.)
Range	0.20–8.80	0.0–7.1	0.0–8.2	0.0–7.7	0.00–49.00	0.0–4.4	77.00–245.00
Average	3.01 ± 2.39	2.8 ± 2.5	1.6 ± 2.3	0.7 ± 1.9	8.38 ± 13.56	1.3 ± 1.2	155.06 ± 51.05

**Table 4 biology-12-00330-t004:** The main marine environmental factors in the waters of the Xuwen coral reef.

Items	Temperature/(°C)	Salinity	pH	DO/(mg/L)	COD/(mg/L)	Suspended Solids /(mg/L)	Transparency /(m)	Oils/(mg/L)
Range	31.75–32.70	33.00–33.24	8.25–8.30	4.04–5.91	0.24–0.88	1.3–31.6	1.2–4.0	0.015–0.043
Average	32.29 ± 0.28	33.15 ± 0.07	8.27 ± 0.01	5.30 ± 0.42	0.49 ± 0.17	17.0 ± 10.5	2.1 ± 0.7	0.024 ± 0.008
Items	NO_3_^−^-N/ (μmol/L)	NO_2_^−^-N / (μmol/L)	NH_4_^+^ -N / (μmol/L)	DIN/ (μmol/L)	PO_4_^3−^-P / (μmol/L)	TP (μmol/L)	TN (μmol/L)	Chla/(µg/L)
Range	0.89–2.89	0.08–1.46	1.13–2.31	2.19–5.31	0.00–0.32	0.22–0.70	4.11–17.46	2.17–19.36
Average	1.79 ± 0.53	0.35 ± 0.34	1.52 ± 0.35	3.66 ± 0.94	0.13 ± 0.11	0.35 ± 0.14	11.69 ± 4.11	7.90 ± 4.79

DO: Dissolved oxygen; COD: Chemical oxygen demand; NO_3_^−^-N: Nitrate-NO_3_^−^-N; NO_2_^−^-: Nitrate- NO_2_^−^-N; NH_4_^+^-N: Ammonium salt; DIN: Dissolved inorganic nitrogen; PO_4_^3−^-P: Inorganic phosphate; TP: Total phosphorus; TN: Total nitrogen; Chla: Chlorophyll-a.

**Table 5 biology-12-00330-t005:** Regression equations between coral community characteristics and environmental factors.

Number	Features of Coral Community	Regression Equation with Environmental Factors
1	Living coral cover	*Y* = 1.913 − 53.431 × NH_4_^+^-N − 0.374 × Chla (*P* _NH4+-N_ = 0.002, *P*_Chla_ = 0.015, *p* < 0.05)
2	Number of coral species	*Y* = 1.311 − 0.310 × Chla + 0.117 × FD (*P*_Chla_ = 0.005, *P*_FD_ = 0.009, *p* < 0.01)
3	Coral diversity index (*H’*)	*Y* = 11.388 − 0.145 × Chla − 0.320 × Dept − 6.869 × Tem + 0.037 × FD (*P*_Chla_ = 0.005, *P*_Dept_ = 0.002, *P*_Tem_ = 0.022, *P*_FD_ = 0.042, *p* < 0.05)
4	Abundance of juvenile corals	*Y* = −1.672 + 0.954 × ID + 0.258 × AH (*P*_ID_ = 0.004, *P*_AH_ = 0.005, *p* < 0.05)
5	Coral bleaching	*Y* = 0.826 − 0.598 × GC (*P*_GC_ = 0.000, *p* < 0.01)

DO: Dissolved oxygen; COD: Chemical oxygen demand; NO_3_^−^-N: Nitrate-NO_3_^−^-N; NO_2_^−^-N: Nitrate- NO_2_^−^-N; NH_4_^+^-N: Ammonium salt; DIN: Dissolved inorganic nitrogen; PO_4_^3−^-P: Inorganic phosphate; Chla: Chlorophyll-a; FD: Fish density; Dept: Depth; Tem: Temperature; ID: Invertebrate density; AC: Macroalgae coverage; AH: Macroalgae height; RC: Reef coverage; GC: Gravel coverage; SC: Sand coverage.

**Table 6 biology-12-00330-t006:** The trend of living coral cover, number of species and dominant groups on the Xuwen coral reef over nearly 10 years.

Years	Number of Species	Living Coral Cover	Dominant Species	References
2004	37	28.9%	*Goniopora gracilis*, *Bernardpora stutchburyi*, *Favites abdita*, *Platygyra daedalea*, *Turbinaria* sp.	Huang et al., 2007 [5], 2004, 18 sites
2008	57	12.1%	*Goniopora planulata*, *Turbinaria peltata*, *Cyphastrea serailia*	Huang et al., 2011 [14], 2008, 25 sites
2020	33	12.6%	*Goniopora gracilis*, *Bernardpora stutchburyi*, *Montipora turgescens*, *Turbinaria peltate*, *Porites lutea*	This study, 2020, 16 sites

## Data Availability

The data presented in this study are available on request from the corresponding author.

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
