# Peer review of "The Ecological Status and Change in High-Latitude Coral Assemblages at the Xuwen Coral Reef, Northern South China Sea: Insight into the Status and Causes in 2020"

_biology, 2023, doi:10.3390/biology12020330_

Round 1

Reviewer 1 Report

Review on Zhenxiong Yang et al., MDPI biology „The ecological status and change in high-latitude coral assemblages at the Xuwen coral reef, northern South China Sea: Insight into the status and causes in 2020”

In their manuscript Zhenxiong Yang et al. present a monitoring study of the Xuwen coral reef in the northern South China Sea. The study comprises ~16 locations sampled with video transects and photo-quadrats on two 50m transects at each location between 2 and 4 m depth. Additionally, water samples at each of the locations were taken and analysed for a wide set of biological, chemical and physical parameters.

As such the outlined sampling provides an update on the status of the Xuwen coral reefs and allows a comparison with previous studies to establish trends in reef development. The results show a similar dominance structure as previous surveys but also a severe decline in reef coverage which the authors attribute to different environmental factors measured during their monitoring campaign.

Obviously the research provides valuable information on trends in reef development, but I have some concerns relating mostly to the applied methods and the interpretation of the gained results.

In my eyes the description of the methods is not complete which makes the interpretation of the data unreliable.

The number of sampling sites is unclear. It is given with 16, but the numbering is from XW1 to XW20 and e.g. in Figure 1 17 sites can be found (and in some other figures as well).

The date and frequency of sampling stay unclear. In general this might not be so relevant for corals as a slow growing organism, but with the indication of a bleaching event during northern summer, this might be of high importance. Also, it is not clear how difficult to identify corals were determined and then sampled (L. 119). A complete list of categories should be given and not an exemplary list L. 124 (e.g. sponges, algae are analysed but not listed). The width of the transect and the size of the quadrats is not indicated.

Furthermore no information is given on dates of sampling of water parameters and how many samples were taken (at each location). As these environmental parameters are analysed in great detail to determine reef reactions a comprehensive description of the sampling strategy is relevant.

In the results section 3.3. information on reef fish are given. However, the methods section gives not indication how these were sampled.

The cited literature originating outside the authors direct geographical neighbourhood seems to be quite outdated in many cases and does not reflect the state of the art (e.g. references 17, 18, 48, 62, 66, 67,68,73) with newer studies being available for most of the cases.

The authors give a lot of emphasis on relating the measured environmental factors with the coverage of corals and their development over time and analyse them in detail. However, no detailed information has been given on the sampling scheme and I assume it was in temporal relation to the sampling of the corals. In this case the measured environmental factors would reflect a tiny fraction of a sometimes highly variable situation, with factors changing with daytime, season, weather conditions and past events if coastal run-offs are considered. Here, the interpretation should be much more careful and should relate to the sampling scheme and eventually to more detailed information on the land side and it’s direct influences. Depending on the sampling scheme this part of the results and the discussion would need considerable rewriting.

Figures are frequently presented with very small legends, so that there are very difficult (impossible) to read in the given size (Figures 1, 2,3, 6b, 8)

The references to figures are not resolved and I would recommend a thorough check for the English as in quite a number of cases sentences are difficult to grasp and sentence parts sometimes have a wrong relation. I did not indicate all occurrences.

Some details:

L. 22 ‘suitability … of reefs’ ?

L. 26 sentence structure (meaning)

L. 28 sustainable management would already include conservation

L. 36 which diversity indicator

L. 37 what do you mean if you say the ‘structure is not stable’

L. 91 question 3 seems a bit unclear

L. 103 the mean annual air temperature differs from the one given in line 106

L. 104. a nearly 20 year old source for climate data might be outdated in current times of global clime change, please update

L. 112 ff. Please be more precise with the sampling scheme (see above). How broad are the transects? What size of quadrats? How many quadrats per site?

L. 116 this is imprecise. Are transects laid at 2 and 4 m depth each or somewhere in-between? Later you analyse the depths separately.

L. 118. as Veron and Zou are two different references the should be written as Veron[24] and Zou [25]

L. 142 do you mean ‘importance’ values?

L. 145 the ‘dominant’ coral?

L. 148 not the ‘sum’

L. 157 Raunkiaer classifies plants according to their life forms (position of buds with respect to the soil surface to survive detrimental conditions (e.g. winter)). Without some very good explanation the use of this concept seems extremely implausible.

L. 185 What kind of ArcGIS methods (Arc GIS is a software). What is the basis for the calculation. What do you define as suitable habitat for corals?

L. 188 Not sure how you can say anything about 5 m depth if 2-4 m depth was sampled?

L. 189 Shuiwei is not on the Map in Fig. 1

L. 193 the retrieved sampling data could be given as a table in the appendix to give the reader an overview.

L. 196 how did you calculate mortality rate (death per time span) with only one sampling per site?

Fig 2b. The bin sizes of the cover categories seem completely arbitrary,

Chapter 2.3 As written, usage of Raunkiaer does not seem plausible (applies also to Figure 4).

L. 234 Satandard → standard

L. 246 Not sure what you mean by a ‘habitable feature’ in this context

L. 247 Methods have to be updated to cover this section

L. 267ff I am getting confused here. XW8 is very, very near to XW16 according to Fig 1, but seems to have very different results (XW16 greatest anthropogenic impact, XW8 highest coverage and species number), XW6/7 suffer from human activity or not (L. 271). XW15 is sometimes included in the study (17th site) but e.g. not listed in Fig. 2 and Fig.3

Figure 6. if environmental factors have been measured just once the validity of the analyses is questionable.

L. 286 not clear what is meant with distribution scope

L288 again: it is not clear how the area can be calculated with the described methods (transects)

L. 291 but please consider that the number of sites in 2008 was 25 (~50% more) and the saturation in species number might not be reached with 16 (or 18) sites

Table. 4 Interestingly the overlap in dominant species with the 2008 study is quite low (only 1 species).

L. 335 warm water corals?

L. 339 bleaching (frequently leading to death) occurs in many place already with temperature lower than 40°C

L.335 please update the methods to indicate which period was sampled. August 2020 does not seem to be part of this study (l.199) Please cite the data for 2021 (and August 2020).

L. 371. Again I am confused. I would expect that the Xuwen reefs are in coastal waters?

Table 5 could be considered to belong into the results section

L. 449 0.567?

L. 451 type of suspended solids?

L. 458 species are different from Fig. 4

L. 513 your results indicate a bleaching percentage of 1.6% in average. This would not be called a mass bleaching event. Thus no severe mortality due to bleaching can occur.

Author Response

The authors would like to thank the reviewers for their time reviewing the manuscript and providing thoughtful comments. We carefully addressed each one of the comments and revised the manuscript accordingly. Point-by-point responses to the reviewers’ comments are presented below. The reviewers’s comments are in black, and the authors’ responses are in red.

Point 1: The number of sampling sites is unclear. It is given with 16, but the numbering is from XW1 to XW20 and e.g. in Figure 1 17 sites can be found (and in some other figures as well).

Response 1:

Thanks for the reviewer’s observation. We have checked the number of samlping sites and revised the related figures. Corrected in the revision.

Point 2: The date and frequency of sampling stay unclear. In general this might not be so relevant for corals as a slow growing organism, but with the indication of a bleaching event during northern summer, this might be of high importance. Also, it is not clear how difficult to identify corals were determined and then sampled (L. 119). A complete list of categories should be given and not an exemplary list L. 124 (e.g. sponges, algae are analysed but not listed). The width of the transect and the size of the quadrats is not indicated. Furthermore no information is given on dates of sampling of water parameters and how many samples were taken (at each location). As these environmental parameters are analysed in great detail to determine reef reactions a comprehensive description of the sampling strategy is relevant.

Response 2:

We agree with the reviewer’s comments and have provided more details for the sampling survey in the revised text:

This field survey was implemented at the Xuwen coral reef in July 2020. 16 sections (XW1-XW20) were set up around the reef. In each section, two transects with measuring tapes (50 m long) were laid parallel to the shoreline at depths between 2 and 4 m (the depths of sections range from 1.5 to 6.7 m, those depths were all covered the area from the upper to lower limits of coral growth in our survey). For fish, the width of the transect each side was 2.5 m; for invertebrate, the width of the transect each side was 1 m. The 10 quadrats (0.5m × 0.5m) were surveyed on each transect. Four photographs were taken for each quadrat, each occupying one-quarter of the quadrat. There were 23 transects and 230 quadrats in our survey.

In our survey, we measured the number of 50 m long transect at each 10 cm scale for corals (Within each radial transect all corals ≥5 cm in diameter were identified to species (or genus for some smaller corals)), and mortality rate of coral was calculated the percentage of dead corals within one year. We extracted data such as sediment type, scleractinian coral assemblage, abundance of juvenile corals, density of coral reef fish, and macroalgae cover from the video transects and quadrats.

In addition, sampling was performed in 16 water sites, including surface (0.5 m) and bottom (Water layer 2m from the bottom) layers. The sampling of water parameters were Temperature(T),salinity(S), pH, transparency, suspended solids (SS), dissolved oxygen(DO), chemical oxygen demand(COD), dissolved inorganic nitrogen(DIN,e.g.,NH4+-N,NO2--N,NO3--N), inorganic phosphate(PO43--P), total nitrogen- (TN), total phosphorus(TP), oils and Chla. There were 405 water samples in our survey.

We have revised a complete list of categories in Table 1. In this study, only coral, fish and algae are identified to species (or genus).

Point 3: In the results section 3.3. information on reef fish are given. However, the methods section gives not indication how these were sampled.

Response 3:

Corrected in the revision. See above Response 2. For fish, the width of the transect each side was 2.5 m.

Point 4: The cited literature originating outside the authors direct geographical neighbourhood seems to be quite outdated in many cases and does not reflect the state of the art (e.g. references 17, 18, 48, 62, 66, 67,68,73) with newer studies being available for most of the cases.

Response 4:

Corrected in the revision. Following the suggestion, we have cited more newer publications and reviewed as follows:

17.Hughes, T.P; Graham, N.A.; Jackson, J.B.; Mumby, P.J.; Steneck, R.S. Rising to the challenge of sustaining coral reef resilience. Trends Ecol Evol. 2010, 25, 633–642. [CrossRef]

18.Yu, K.F. Coral reefs in the South China Sea: their response to and records on past environmental changes. Sci China Earth Sci. 2012, 55, 1217–1229. [CrossRef]  

19.De’ath, G.; Fabricius, K.E.; Sweatman, H.; Puotinen, M. The 27-year decline of coral cover on the Great Barrier Reef and its causes. Proc Nat Acad Sci USA .2012,109. 17995–17999. [CrossRef]

23.Pandolfi, J.M.; Roger, H.B.; Enric, S.; Terence, P. H.; Karen, A. B.; Richard, G.C.; Deborah, M.; Loren, M.; Marah, J.H.N.; Gustavo, P.; Robert, R.W.; Jeremy, B.C.J. Global trajectories of the long-term decline of coral reef ecosystems. Science. 2003, 301. 955–958. [CrossRef]

27.National Meteorological Centre. Xuwen weather forecast. 2022. [CrossRef]

43.Mehmood, A.; Shah, A.H.; Shah, A.H.; Khan, S.U.; Khan, K.R. Farooq, M.; Ahmad, H.; Sakhi, S. Classification and ordination analysis of herbaceous flora in district Tor Ghar, western Himalaya. Acta. Ecol. Sin. 2021,41(5). 451-462. [CrossRef]

  1. Jordan, L.K.B.; Banks, K.W.; Fisher, L.E.; Walker, B.K.; Gilliam, D.S. Elevated sedimentation on coral reefs adjacent to a beach nourishment project. Mar. Pollut. Bull. 2010, 60:261–271. [CrossRef]
  2. Schiel, D.R.; Wood, S.A.; Dunmore, R.A.; Taylor, D.I. 2006. Sediment on rocky intertidal reefs: effects on early post-settlement stages of habitat-forming seaweeds. J. Exp. Mar. Biol. Ecol. 2006, 331, 158–172. [CrossRef]

81.Thurber, R.L.V.; Burkepile, D.E.; Fuchs, C.; Shantz, A.A.; McMinds, R.; Zaneveld, J.R. Chronic nutrient enrichment increases prevalence and severity of coral disease and bleaching. Glob. Change. Biol. 2014, 20.544–554. [CrossRef]

Point 5: The authors give a lot of emphasis on relating the measured environmental factors with the coverage of corals and their development over time and analyse them in detail. However, no detailed information has been given on the sampling scheme and I assume it was in temporal relation to the sampling of the corals. In this case the measured environmental factors would reflect a tiny fraction of a sometimes highly variable situation, with factors changing with daytime, season, weather conditions and past events if coastal run-offs are considered. Here, the interpretation should be much more careful and should relate to the sampling scheme and eventually to more detailed information on the land side and it’s direct influences. Depending on the sampling scheme this part of the results and the discussion would need considerable rewriting.

Response 5:

We thank the reviewer for the advice. We have added a more detailed description and discussion in the revised manuscript. For the sampling scheme, we have added detailed information of the sampling methodology. In the discussion section, we have revised some details about the evaluation of the status and causes in the manuscript, and the information not very relevant to the objectives has been removed and the objectives of the paper have been emphasized.

As reef corals are very sensitive to subtle changes of environmental conditions, they are widely used in the monitoring of seawater quality. Although some ecology of scleractinian corals has been studied in Xuwen before this work, with the rapid development of the economy and society, the influence of anthropogenic activities (e.g., inshore aquaculture, coastal engineering) is more frequent than in the past. Since 2010, long-term monitoring has been carried out in Xuwen coral reef, and monitoring time is mainly in summer. However, those survey sites are inconsistent and environmental parameters are different, which make it difficult to quantify and evaluate the impact of anthropogenic factors. In order to learn about the spatial distribution pattern and ecological status of coral reef resources in the South China Sea today, we carried out a series of surveys starting in 2016, including the Xuwen coral reef.

Xuwen coral reef is nearby the Qiongzhou Strait, there is no large land-based runoff to the Xuwen coral reef, significantly influenced by currents of Qiongzhou Strait. Our monitoring data (unpublished data) show that there is little difference in seawater quality. For instance, in summer, from July 2010 to July 2020, the mean annual water temperature ranges from 30.20 to 32.29°C. The survey in July 2020 was a large-scale ecological survey implemented in the Xuwen coral reef recently. Our survey is not only for water environment, but also for the anthropogenic activities. There are a large number of fixed nets and waste fishing nets have been found in our survey. Human activities nearshore, such as marine culture and fishing, are frequent in the Xuwen coral reef. Indeed, damage from extreme weather, such as typhoons, can directly damage reefs and cause coral mortality. However, if the ecological environment is good, the coral can recover naturally without the disturbance of human activities. For instance, there are two typhoons (Super Typhoon Rammasun, No. 1409; Typhoon Kalmaegi, No.1415) struck Xuwen coral reef in 2014, and the mean living coral cover in 2015 was 15.1% [reference 15], however in 2018, the mean coral cover was 8.96% [reference 16], it seems to indicate that the coral did not recover naturally in 2018. Hence, we speculat that, in addition to extreme weather damage such as typhoons, the changes in the Xuwen coral community in recent years are significantly related to the influence of human activities. In addition, our results show that the coral community in 2020 might be greatly affected by high suspended solids and elevated nutrient levels, which are related to the current, mariculture and coastal erosion in and around Xuwen. Our monitoring results for seawater quality and human activities are consistent.

Point 6: Figures are frequently presented with very small legends, so that there are very difficult (impossible) to read in the given size (Figures 1, 2,3, 6b, 8)).

Response 6:

Corrected in the revision.

Point 7: The references to figures are not resolved and I would recommend a thorough check for the English as in quite a number of cases sentences are difficult to grasp and sentence parts sometimes have a wrong relation. I did not indicate all occurrences.

Response 7:

As suggested, we have carefully revised the text to be clear, including the structure, grammer, English expression, and so on.

Point 8: L. 22 ‘suitability … of reefs’?).

Response 8:

Removed in the revision.

Point 9: L. 26 sentence structure (meaning).

Response 9:

Corrected in the revision.

Point 10: L. 28 sustainable management would already include conservation.

Response 10:

Corrected in the revision.

Point 11: L. 36 which diversity indicator).

Response 11:

Corrected in the revision. The diversity indicator is Shannon-Wiener index(H’).

Point 12: L. 37 what do you mean if you say the ‘structure is not stable’.

Response 12:

By this we mean that coral community was uneven, and there was a trend of population differentiation and succession. We removed the use of Raunkiaer classification and therefore we removed ‘structure is not stable’ in the revision.

Point 13: L. 91 question 3 seems a bit unclear.

Response 13:

Corrected in the revision. We have revised the question clear, that is, what is the cause of those environmental factors?

Point 14: L. 103 the mean annual air temperature differs from the one given in line 106.

Response 14:

Corrected in the revision.

Point 15: L. 104. a nearly 20 year old source for climate data might be outdated in current times of global clime change, please update.

Response 15:

Corrected in the revision. We have updated newer climate data.

Point 16: L. 112 ff. Please be more precise with the sampling scheme (see above). How broad are the transects? What size of quadrats? How many quadrats per site?.

Response 16:

Corrected in the revision. See above (Response 2).

Point 17: L. 116 this is imprecise. Are transects laid at 2 and 4 m depth each or somewhere in-between? Later you analyse the depths separately.

Response 17:

Corrected in the revision. See above (Response 2).

Point 18: L. 118. as Veron and Zou are two different references the should be written as Veron[24] and Zou [25].

Response 18:

Corrected in the revision.

Point 19: L. 142 do you mean ‘importance’ values?.

Response 19:

Yes, we did. Corrected in the revision.

Point 20: L. 145 the ‘dominant’ coral?.

Response 20:

Corrected in the revision. We revised dominant species of corals.

Point 21: L. 148 not the ‘sum’.

Response 21:

Corrected in the revision.

Point 22: L. 157 Raunkiaer classifies plants according to their life forms (position of buds with respect to the soil surface to survive detrimental conditions (e.g. winter)). Without some very good explanation the use of this concept seems extremely implausible.

Response 22:

Removed in the revision. We thank the reviewer for the advice. we removed the use of Raunkiaer classification.

Point 23: L. 185 What kind of ArcGIS methods (Arc GIS is a software). What is the basis for the calculation. What do you define as suitable habitat for corals?.

Response 23:

Thanks for the reviewer’s advice. We have corrected in the revision. The area is calculated by the geometry area calculation function of ArcMap software, using China Geodetic Coordinate System 2000, with Gauss Kruger 3 Degree projection, the central meridian is 111°E. The concept of suitable habitat is defined as the area from the upper to lower limits of coral growth by artificial diving in our survey.

Point 24: L. 188 Not sure how you can say anything about 5 m depth if 2-4 m depth was sampled?

Response 24:

We have revised it for clarity. The actual water depth at each site is not only 4 meters. In each section, two transects with measuring tapes (50 m long) were laid parallel to the shoreline at depths between 2 and 4 m (the depths of sections range from 1.5 to 6.7 m, those depths were all covered the area from the upper to lower limits of coral growth in our survey).

Point 25: L. 189 Shuiwei is not on the Map in Fig. 1.

Response 25:

Corrected in the revision.

Point 26: L. 193 the retrieved sampling data could be given as a table in the appendix to give the reader an overview.

Response 26:

Corrected in the revision. We thank the reviewer for the advice. We have added a sampling data table for main reel dwelling organism in Table 3.

Point 27: L. 196 how did you calculate mortality rate (death per time span) with only one sampling per site?

Response 27:

Corrected in the revision. The coral death is the dead coral within one year in this study. In our survey, we measured the number of 50 m long transect at each 10 cm scale for corals (Within each radial transect all corals ≥5 cm in diameter were identified to species (or genus for some smaller corals)), and mortality rate of coral was calculated the percentage of dead corals within one year on each transect.

Point 28: Fig 2b. The bin sizes of the cover categories seem completely arbitrary.

Response 28:

Corrected in the revision. We thank the reviewer for the advice. The bin sizes of cover categories are classified by ArcMap software, Using Natural Breaks (Jenks) method.

Point 29: Chapter 2.3 As written, usage of Raunkiaer does not seem plausible (applies also to Figure 4).

Response 29:

Removed in the revision. We thank the reviewer for the advice. After considering carefully, we removed the use of Raunkiaer classification.

Point 30: L. 234 Satandard → standard.

Response 30:

Removed in the revision.

Point 31: L. 246 Not sure what you mean by a ‘habitable feature’ in this context.

Response 31:

Corrected in the revision. We have revised those concepts in the result section. What we mean is reef dwelling organism composition and environmental factors in water.

Point 32: L. 247 Methods have to be updated to cover this section.

Response 32:

Corrected in the revision. See above (Response 2).

Point 33: L. 267ff I am getting confused here. XW8 is very, very near to XW16 according to Fig 1, but seems to have very different results (XW16 greatest anthropogenic impact, XW8 highest coverage and species number), XW6/7 suffer from human activity or not (L. 271). XW15 is sometimes included in the study (17th site) but e.g. not listed in Fig. 2 and Fig.3.

Response 33:

Corrected in the revision. Following the suggestions, we have made a more detailed description and discussion about these 2 sites. It should be noted that XW8 site was closely to XW6 site, where it’s concentration of suspended solids was 31.5 mg/L, however corals were growing well there. The water environment at site XW16 was the same as site XW8, but the concentration of Chla was higher than site XW8 (XW16, 4.58 µg/L; XW8, 8.69 µg/L). In addition, XW16 site had a higher sand coverage than XW8 site. Generally, the high Chla content indicates high phytoplankton biomass, indicating that there was a high nutrient content in the water. We suspected the high Chla content and sand coverage in XW16 site might indicate there is not a suitable environment for coral growth or recovery. More evidence is needed for further research in the future.

Point 34: Figure 6. if environmental factors have been measured just once the validity of the analyses is questionable.

Response 34:

Agreed. We have responsed to those in Response 5.

As mentioned above, the survey in July 2020 was a large-scale ecological survey implemented in the Xuwen coral reef recently. Our survey is not only for water environment, but also for the anthropogenic activities. In this manuscript, based on the data acquired by investigation and the implementation of statistical analysis methods (e.g. DCA, PCA, RDA), potential environmental influence factors for coral communities in Xuwen were proposed. Those methods need to consistent sampling data of sites. However, the historical survey sites are inconsistent and environmental parameters are different, which make it difficult to quantify and evaluate the impact of anthropogenic factors. We cannot use those historical data for analyzing potential environmental influence factors. Fortunately, our monitoring data show that there is little difference in seawater quality. Hence, considering only from the water environment monitoring, our results could reflect the state of the Xuwen coral reef in the short run. To learn about the spatial distribution pattern and ecological status of Xuwen coral reef, we will implement these long-term monitoring in the same locations in the future as in 2020.

Point 35: L. 286 not clear what is meant with distribution scope.

Response 35:

Removed in the revision. We thank the reviewer for pointing out the contradiction.

Point 36: L288 again: it is not clear how the area can be calculated with the described methods (transects).

Response 36:

Corrected in the revision. See above (Response 2).

Point 37: L. 291 but please consider that the number of sites in 2008 was 25 (~50% more) and the saturation in species number might not be reached with 16 (or 18) sites.

Response 37:

Corrected in the revision. We thank the reviewer for the advice. For corals that were difficult to identify in the field, sampling and identification were carried out in the laboratory. Each person has his or her own identification opinion, which may make the number of species in the same site different. In order to make the identification more reasonable, we have referred to the historical study of Xuwen coral reef.

Point 38: Table. 4 Interestingly the overlap in dominant species with the 2008 study is quite low (only 1 species).

Response 38:

We thank the reviewer for pointing out this finding. In order to make the identification more reasonable, we have referred to historical studies in Xuwen coral reef. Our results regarding the dominant coral groups are the same as the results of historical studies, that is, the dominant groups are Poritidae, Merulinidae, and Dendrophylliidae. In addition, Goniopora sp. (For corals, it is difficult to identify to species, in particular for genus Goniopora) is the dominant specie from 2004 to 2020. The dominant species with the 2008 study are still belong to family Poritidae, Merulinidae, and Dendrophylliidae. Considering from the perspective of genera, we believe that the dominant species of Xuwen coral reef has not changed.

Point 39: L. 335 warm water corals?.

Response 39:

We have corrected them as suggested.

Point 40: L. 339 bleaching (frequently leading to death) occurs in many place already with temperature lower than 40°C.

Response 40:

Removed in the revision.

Point 41: L.335 please update the methods to indicate which period was sampled. August 2020 does not seem to be part of this study (l.199) Please cite the data for 2021 (and August 2020).

Response 41:

Corrected in the revision. We thank the reviewer for the advice. We have made a more detailed description and discussion about this section.

From July 2020 to July 2021, we found a mass coral bleaching event on Xuwen. In July 2020 (16 sites, the mean mortality rate of coral in one year was 2.8% and the mean living coral cover was 12.1%, this study), 62.5% of sites experienced coral bleaching, with 1.6% mean bleaching. In August 2020, a mass coral bleaching event had happened in NSCS, our monitoring data showed that (10 sites in Xuwen, 26 transects), mean coral bleaching reached 89.3% (all sites had been found coral bleaching, range from 68.0% to 100.0%) with increasing sea temperature. However, in the later period of monitoring, as the sea temperature fell, coral bleaching decreased, with a mean of 5.8% in July 2021 (17 sites, the mean mortality rate of coral in one year was 0.33% and the mean living coral cover was 11.7%, unpublished data).

Point 42: L. 371. Again I am confused. I would expect that the Xuwen reefs are in coastal waters?

Response 42:

We thank the reviewer for pointing out the problem. Corrected in the revision, more detailed description and discussion were performed about this section.

Xuwen coral reef is nearby the Qiongzhou Strait, there is no large land-based runoff to the Xuwen coral reef, High influenced by tidal current of Qiongzhou Strait. Unlike the coastal water with high temperature and low salt in other area, the water temperature of Xuwen coral reefs in summer was little different, that is, no too high in summer. The mean annual water temperature ranges from 30.20 to 32.29 °C in last ten years in summer. Zhao et al. (2008, 2002) estimated that the Qiongzhou Strait tidal current was strong and that the water was exchanging and mixing frequently between Beibu bay and the NSCS, which could make the sea temperature cool in the summer. This may be one reason why corals in the Xuwen coral reefs were still growing or recovering even though the sea temperature was high for a period of time in 2020.

Zhao, H.T.; Wang, L.R.; Song, C.J. conditions for the existence and development of coral reefs on the West Bank of Xuwen County. Trop. Geo. 2008, 5, 234-241. [CrossRef]

Zhao, H.T.; Wang, L.R.; Song, C.J., Yu, K.F.; Yuan, J.Y. Features of fringing reef at Dengloujiao, Leizhou Peninsula. Mar. Geol. Quat. Geol. 2002, 22(2), 35-40.

Point 43: Table 5 could be considered to belong into the results section.

Response 43:

Corrected in the revision.

Point 44: L. 449 0.567?.

Response 44:

Corrected in the revision.

Point 45: L. 451 type of suspended solids?

Response 45:

Corrected in the revision. Suspended solids in water.

Point 46: L. 458 species are different from Fig. 4.

Response 46:

Corrected in the revision.

Point 47: L. 513 your results indicate a bleaching percentage of 1.6% in average. This would not be called a mass bleaching event. Thus no severe mortality due to bleaching can occur.

Response 47:

Corrected in the revision. We thank the reviewer for the advice. We have made a more detailed description and discussion about this section. See above (Response 41). Lyu et al. (2022) have reported this coral bleaching event in NSCS from August to September 2020.

Lyu, Y.H.; Zhou, Z.H.; Zhang, Y.M. Chen, Z.Q.; Deng, W.; Shi, R.G. The mass coral bleaching event of inshore corals form South China Sea witnessed in 2020: insight into the causes, process and consequence. Coral Reefs. Report. 2022, Published online. [CrossRef]

Reviewer 2 Report

The study on ‘The ecological status and change in high-latitude coral assemblages at the Xuwen coral reef, northern South China Sea: Insight into the status and causes in 2020’ by Yang et al., has investigated the ecological features of the scleractinian coral assemblage and their correlation with environmental factors from the Xuwen coral reef. To me, it’s an interesting study and the MS is well-structured. However, the manuscript needs some minor revisions before considering for final publication:

1. Materials and Methods: For corals that were difficult to identify in the field, sampling and identification were carried out in the laboratory - please use reference.

2. Page-4, line 151: Do you mean species diversity (Shannon-Wiener, H′), evenness (Pielou’s, J′) and species richness (Margalef, D)?

3. Page-4, line 155: S is the total number of species; and N is the total number of individuals-right?

4. Page-4: Please consider using the following formula for expressing the diversity indices in page 4

H′ = –       OR    H = -∑[(pi) × ln(pi)]

 J′ = H′ / lnS

D = (S – 1) / lnN

5. Page 5, line 177: Canoco 4.56. Please add a reference for this software.

7. Please check the family and species names carefully throughout the MS.

8. Page 11, line 286: Too many citations, please reduce some.

9. The discussion part is too long, may you consider rewriting it.

Overall, good structured MS, and enjoyed reading the contents.

Author Response

The authors would like to thank the reviewers for their time reviewing the manuscript and providing thoughtful comments. We carefully addressed each one of the comments and revised the manuscript accordingly. Point-by-point responses to the reviewers’ comments are presented below. The reviewers’s comments are in black, and the authors’ responses are in red.

Point 1: Materials and Methods: For corals that were difficult to identify in the field, sampling and identification were carried out in the laboratory - please use reference.

Response 1:

We thank the reviewer for the advice. Corrected in the revision. We have added the reference as suggested. To reduce identification errors, we have referred to historical studies in Xuwen coral reef, and species identification was mainly referred to the study of Huang et al.

  1. Huang, H.; Lian, J.S.; Wang, H.J.; Chen, Y.H. Xuwen coral reefs and their biodiversity. Ocean Press, Beijing, China, 2007.7.

Point 2: Page-4, line 151: Do you mean species diversity (Shannon-Wiener, H′), evenness (Pielou’s, J′) and species richness (Margalef, D)?.

Response 2:

We thank the reviewer for the advice.Corrected in the revision. We mean species diversity (Shannon-Wiener, H’), evenness (Pielou’s, J’) and species richness (Margalef, D).

Point 3: Page-4, line 155: S is the total number of species; and N is the total number of individuals-right?

Response 3:

Right. We thank the reviewer for the advice. Corrected in the revision as suggested.

Point 4: Page-4: Please consider using the following formula for expressing the diversity indices in page 4.

H= - OR H =

J’=H/lnS 

D=(S-1)/lnN

Response 4:

We thank the reviewer for the advice. Corrected in the revision as suggested.

Point 5: Page 5, line 177: Canoco 4.56. Please add a reference for this software.

Response 5:

We thank the reviewer for the advice. Corrected in the revision. We have added the reference as suggested.

43.Mehmood, A.; Shah, A.H.; Shah, A.H.; Khan, S.U.; Khan, K.R. Farooq, M.; Ahmad, H.; Sakhi, S. Classification and ordination analysis of herbaceous flora in district Tor Ghar, western Himalaya. Acta. Ecol. Sin. 2021,41(5). 451-462. [CrossRef]

Point 6: Please check the family and species names carefully throughout the MS.

Response 6:

We thank the reviewer for the advice. Corrected in the revision.

Point 7: Page 11, line 286: Too many citations, please reduce some.

Response 6:

We thank the reviewer for the advice. Corrected in the revision.

Point 8: The discussion part is too long, may you consider rewriting it.

Response 8:

We thank the reviewer for the advice. Corrected in the revision. We have revised the discussion and results part. For instance, the analysis of correlation and multiple linear regressions between coral community characteristics and environmental factors have revised to ‘Result’ section. More sampling data and categories of species have been added to ‘Result’ section too. Generally, we have made a more detailed description and discussion about this MS. 

Round 2

Reviewer 1 Report

The authors have greatly improved the manuscript and addressed many of my previous comments and concerns. However, some issues remains and are listed below. Furthermore, I saw that there are still some issues with the English language. For time reasons I did not list these. Sorry! Please check again thoroughly also the new parts.

L.26  potential environmental factors for coral communities in Xuwen were proposed ->  potential environmental conditions favouring coral communities in Xuwen were summarised

L. 91 frequent -> intense          (as continuos processes are listed)

L. 121 if you have 16 field site it is irritating if you refer to them as XW1- XW20. Please be more precise e.g. XW1-XW14, XW16, XW20

L. 129 those depths were all covered ...->  those depths covered...

L.130 how were fish recorded?

L.131  The 10 quadrats -> 10 quadrats...

L.135 Identification of corals still unclear. You took photos and identified corals from them. If corals were difficult to identify you went back to the field to sample them? Or did you do that directly?

L.143 you sampled in July 2020. This does not allow to calculate a mortality rate over 1 year. For this you would need a comparison to a sampling in July 2019. Here you can only give a percentage of dead corals. If you refer to previous data please cite them.

L. 148 not sure what 'artificial diving' is

L. 156 how often did you repeat each sampling?

L.208 reference to table 1?

Figure 2b polygons do not seem to be filled correctly, 2a the dots have different sizes than in the legend, seem okay for 2b

L.226 please indicate what you correlated in the text
L.250 mortality rate cannot be given with the describe sampling design

Table 3 no mortality rate can be given

L.291 what does etc refer to here?

L.341 Not sure how you can come to this conclusion. In the methods you describe that coral area is calculated as an oceanographic feature (area between min and max depth corals can occur). It is unlikely that this changes during the years. If you extrapolate life coral cover to the whole region, you have to describe the method as well.

Author Response

The authors would like to thank the reviewers for their time reviewing the manuscript and providing thoughtful comments. We carefully addressed each one of the comments and revised the manuscript accordingly. Point-by-point responses to the reviewers’ comments are presented below. 

Point 1: I saw that there are still some issues with the English language. For time reasons I did not list these. Sorry! Please check again thoroughly also the new parts.

Response 1:

Corrected in the revision. As suggested, we have carefully revised the text.

Point 2: L.26 potential environmental factors for coral communities in Xuwen were proposed -> potential environmental conditions favouring coral communities in Xuwen were summarised.

Response 2:

Corrected in the revision.

Point 3:  frequent -> intense (as continuos processes are listed).

Response 3:

Corrected in the revision.

Point 4: L. 121 if you have 16 field site it is irritating if you refer to them as XW1- XW20. Please be more precise e.g., XW1-XW14, XW16, XW20.

Response 4:

We thank the reviewer for the advice. We have revised in the manuscript. We followed this suggestion and revised as: “Based on GPS positioning, 16 sections (XW1-XW14, XW 16, XW20, the locations of all the sampling sites were shown in Figure 1) were set up around the reef.”

Point 5: L. 129 those depths were all covered ...-> those depths covered...

Response 5:

Corrected in the revision.

Point 6: L.130 how were fish recorded?

Response 6:

Corrected in the revision. We recorded the number and species of fish through video and close-up photographs.

Point 7: L.131 The 10 quadrats -> 10 quadrats...

Response 7:

Corrected in the revision.

Point 8: L.135 Identification of corals still unclear. You took photos and identified corals from them. If corals were difficult to identify you went back to the field to sample them? Or did you do that directly?

Response 8:

We agree with the reviewer’s comments. We have revised in the manuscript. In our study, we recorded the number and species of coral through video and close-up photographs of the survey directly. In 50 m long transect at each 10 cm scale for corals. Corals within each radial transect all corals ≥5 cm in diameter were identified to species (or genus for some smaller corals), including alive, bleached and recently dead corals. Mortality rate of coral was calculated the percentage of recently dead corals (colonies entirely covered by turfing algae, but with discernible skeletal structure). In our study, named species in Xuwen coral reef mainly referred to the study of Huang et al. (e.g., references 5, 14), because they have carried out lots of research works in Xuwen coral reef in the past. In addition, the corals in this survey are all common species, which can be identified from video and close-up photos directly. if the law allowed, we will collect coral samples for skeletal identification.

The references were cited as follows:

  1. Huang, H.; Lian, J.S.; Wang, H.J.; Chen, Y.H. Xuwen coral reefs and their biodiversity. Ocean Press, Beijing, China,7.
  2. Huang, H.; Zhang, Y.Y.; Lian, J.S.; Li, X.B.; You, F.; Yang, J.H.; Lei, X.M.; Zhang, C.L. Structure and diversity of scleractinia coral communities along the west seashore of Xuwen County. Biodiversity Sci. 2011, 19(5), 505-510.

Point 9: L.143 you sampled in July 2020. This does not allow to calculate a mortality rate over 1 year. For this you would need a comparison to a sampling in July 2019. Here you can only give a percentage of dead corals. If you refer to previous data please cite them.

Response 9:

We agree with the reviewer’s comments. We have added more details in the revision. In our study, mortality rate of coral was the percentage of recently dead corals. We define recently dead corals by skeletal structure of colonies and degree of algal cover on colonies. According to Huang et al. (2012) and Depczynski et al. (2013) reporting, recently dead corals is colonies entirely covered by turfing algae, but with discernible skeletal structure. 

The references were cited as follows:

â‘ Huang, H.; You, F.; Lian, J.S.; Zhang, C.L.; Yang, J.H.; Li, X.B.; Yuan, T.; Dong, Z.J. Composition and distribution of scleractinian coral in the northwest of Hainan island. Marine Sciences. 2012, 36(9), 64-74

â‘¡Depczynski, M.; Gilmour, J. P.; Ridgway, T.; Barnes,H.; Heyward, A. J.; Holmes,T. H.; Moore, J. A. Y.; Radford, B. T.; Thomson, D. P.; Tinkler, P.; Wilson, S. K. Bleaching, coral mortality and subsequent survivorship on a West Australian fringing reef. Coral Reefs. 2013,32,233–238. DOI 10.1007/s00338-012-0974-0

â‘¢State Oceanic Administration. Technical specification for eco-monitoring of coral reef ecosystem (HY/T 082-2005), Standards Press of China, Beijing, China, 2005.

Point 10: L. 148 not sure what 'artificial diving' is

Response 10:

Corrected in the revision. What we mean is scuba diving, the divers entered into the water and record status of the reefs.

Point 11: L. 156 how often did you repeat each sampling?

Response 11:

Once sampling in each site. Corrected in the revision.

Point 12: L.208 reference to table 1?

Response 12:

Corrected in the revision. We followed this suggestion and revised as: “The records of species at the Xuwen Coral are shown in Table 1.”

Point 13: Figure 2b polygons do not seem to be filled correctly, 2a the dots have different sizes than in the legend, seem okay for 2b.

Response 13:

Corrected in the revision. Figure 2b polygons have been refilled, 2a the dots have been modified to the same size as in the legend. 

Point 14: L.226 please indicate what you correlated in the text.

Response 14:

Corrected in the revision. Our results show correlations between the dominant species were no significant differences (P > 0.05), indicating that the interspecific association of the coral community was relatively loose.

We followed this suggestion and revised as: “The spearman rank correlation between the dominant species in coral communities is shown in Figure 3. The number of positive correlations (46, accounts for 59.0%) between the dominant species was slightly higher than those of negative correlations (32, accounts for 41.0%), and the number of significant correlations (P<0.05) was 6, accounts for 7.7%, indicating that the interspecific association of the coral community was relatively loose.”

Point 15: L.250 mortality rate cannot be given with the describe sampling design, Table 3 no mortality rate can be given.

Response 15:

We agree with the reviewer’s comments. We have responded above. Corrected in the revision.

Point 16: L.291 what does etc refer to here?

Response 16:

Removed.

Point 17: L.341 Not sure how you can come to this conclusion. In the methods you describe that coral area is calculated as an oceanographic feature (area between min and max depth corals can occur). It is unlikely that this changes during the years. If you extrapolate life coral cover to the whole region, you have to describe the method as well.

Response 17:

Agreed. Following the suggestions, we have removed the comparison of coral area and revised as: “Past research [5-6,8-16] showed that the living coral cover of the Xuwen coral reef had been greatly reduced in recent years, which was reduced by 56.4% compared to that in 2004 (Table 6).”
